# DiSa: Saliency-Aware Foreground-Background Disentangled Framework for Open-Vocabulary Semantic Segmentation

## Abstract

Open-vocabulary semantic segmentation aims to assign labels to every pixel in an image based on text labels. Existing approaches typically utilize Vision-Language Models (VLMs), such as CLIP, for dense prediction. However, VLMs, pre-trained on image-text pairs, are biased toward salient, object-centric regions and exhibit two critical limitations when adapted to segmentation: (i) *Foreground Bias*, which tends to ignore background regions, and (ii) *Limited Spatial Localization*, resulting in blurred object boundaries. To address these limitations, we introduce **DiSa**, a novel saliency-aware foreground-background disentangled framework. By explicitly incorporating saliency cues in our designed Saliency-aware Disentanglement Module (SDM), DiSa separately models foreground and background ensemble features in a divide-and-conquer manner. Additionally, we propose a Hierarchical Refinement Module (HRM) that leverages pixel-wise spatial contexts and enables channel-wise feature refinement through multi-level updates. Extensive experiments on six benchmark open-vocabulary semantic segmentation datasets demonstrate that DiSa consistently outperforms current state-of-the-art methods.

## 1 Introduction

Open-vocabulary semantic segmentation aims to label each pixel with an unlimited range of categories that extend beyond a pre-defined closed set, based on text labels. To this end, vision-language models (VLMs), e.g., CLIP (Radford et al., 2021) and ALIGN (Jia et al., 2021), have been widely explored, as they exhibit powerful zero-shot recognition capabilities via large-scale training on image-text pairs.

Despite these advances, VLMs pre-trained on image-text pairs face 2 critical limitations when adapted to dense prediction tasks: (1) *Foreground Bias*: VLMs tend to overemphasize salient, foreground regions while neglecting background context, leading to misclassification of background regions (Li et al., 2024b). This bias stems from the object-centric nature of pre-training data, where captions predominantly describe salient, foreground instances. This results in a fundamental misalignment between the foreground-centric bias of VLMs and the pixel-level precision of segmentation, which requires holistic scene understanding and accurate recognition of non-salient background regions. As shown in row 1 of Fig. 1, VLMs pay little attention to non-salient, background buildings. (2) *Limited Spatial Localization*: VLMs demonstrate limited capacity of fine-grained spatial reasoning required for segmentation predictions. Due to insufficient dense supervision during pre-training, these models struggle to capture precise object boundaries and reconstruct local structural details (as shown in row 2 of Fig. 1). This poses challenges in distinguishing visually similar or spatially overlapping categories (Lee et al., 2025), particularly for small objects and background regions that require nuanced spatial reasoning for accurate segmentation (Zhou et al., 2022; Zhong et al., 2022).

To address these limitations, we propose foreground-background disentanglement mechanisms to tackle various category roles across different visual contexts. Our design is motivated by the observation that most categories exhibit context-dependent roles, e.g., cars or furniture may appear as either foreground instances or background context depending on scene composition. This contextual distinction highlights the importance of adaptive representation learning that captures both fine-grained localization for foreground instances and the semantic coherence for background regions.

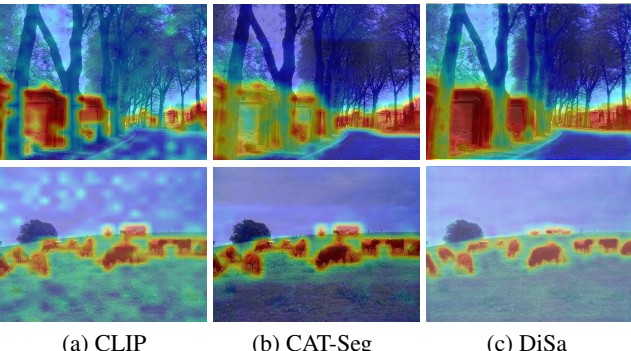

(a) CLIP        (b) CAT-Seg        (c) DiSa

Figure 1: **Visualization of correlation maps.** VLMs face *Foreground Bias* and *Limited Spatial Localization* limitations. Our proposed DiSa effectively alleviates these challenges. The first row indicates class "building", and the second row indicates class "animal".

While existing approaches explore token-level or class-level disentanglement, they either fail to preserve intra-class relationships or assume rigid taxonomies, leading to sub-optimal performance. To overcome these limitations, we leverage saliency cues for adaptive foreground-background inter-token decomposition for each category. Unlike prior works that merely employ saliency for computational efficiency (Choi et al., 2024; Luo et al., 2024), our method explicitly leverages saliency to address the aforementioned *Foreground Bias* challenge. Specifically, we leverage saliency maps derived from text-image cross-attention to effectively partitions per-class visual embeddings into foreground (salient, object-centric) and background (contextual, peripheral) regions based on their corresponding saliency scores. This enables ensemble feature learning via foreground/background dual branches that capture domain-specific characteristics while preserving semantic coherence.

Building on the saliency-aware disentanglement, we propose a novel framework, DiSa, which explicitly separates foreground and background features. This decomposition enables our model to learn distinct and complementary representations, effectively addressing the inherent imbalance in semantic granularity between foreground and background regions. In addition, we introduce a Hierarchical Refinement Module (HRM) that captures detailed spatial context and refines features via multi-level updates. Specifically, it consists of (1) Pixel-wise Refinement, which enhances spatial localization at the pixel level; (2) Category-wise Refinement, which captures channel-wise coherence for each class; and (3) Semantic-wise Refinement, which extracts semantic consistency within broader foreground/background groupings.

In summary, our contributions in this paper include:

- We propose **DiSa**, a novel Saliency-aware Foreground-background Disentangled framework for open-vocabulary semantic segmentation. Our Saliency-aware Disentanglement Module (SDM) is the first to use explicit saliency cues for adaptive intra-class foreground-background decomposition, enabling context-dependent foreground-background assignment. It facilitates semantic coherence especially for non-salient background regions, mitigating *Foreground Bias*.

- DiSa introduces a Hierarchical Refinement Module (HRM) that captures spatial context through Pixel-, Category-, and Semantic-wise Refinement. By incorporating spatial and channel-level context modeling, HRM alleviates the challenge of *Limited Spatial Localization*, improving fine-grained boundary localization and spatial discrimination capabilities.

- We conduct extensive evaluations across six large-scale open-vocabulary semantic segmentation benchmarks. DiSa consistently outperforms state-of-the-art methods, achieving significant performance gains that demonstrate its effectiveness and robustness.

## 2 RELATED WORK

### 2.1 OPEN-VOCABULARY SEMANTIC SEGMENTATION

With the advance of VLMs, researchers have started to explore their powerful visual-text alignment capabilities to provide semantically rich and aligned multimodal representations in this task. SegCLIP

(Liu et al., 2024) integrates CLIP with Vision Transformers (ViT) (Dosovitskiy et al., 2020) through a semantic group module that aggregates patches with learnable centers. It additionally introduces two auxiliary losses, one is a reconstruction loss for recovering the masked patches, and another is a superpixel-based KL divergence loss. CAT-Seg (Cho et al., 2024) estimates cost volumes from CLIP image-text similarities, followed by spatial and class aggregation to improve localization accuracy. The cost volumes serve as visual groundings for class-specific predictions. SCAN (Liu et al., 2024) presents a Semantic-assisted Calibration Network to mitigate misalignment between visual contents and text semantics by calibrating the mask proposals and reducing domain bias in CLIP. ESC-Net (Lee et al., 2025) leverages CLIP-derived image-text correlations as pseudo-supervision for SAM, generating accurate predictions through the powerful segmentation capabilities of foundation models.

A parallel line of research investigates saliency for computational efficiency. SBAM (Choi et al., 2024) proposes an adaptive masking mechanism based on saliency-driven importance scores to enhance pre-training efficiency. PnP-OVSS (Luo et al., 2024) introduces a novel token pruning strategy that constructs class-agnostic saliency maps by aggregating category-specific attention, gradually pruning less discriminative tokens. However, these approaches merely focus on computational optimization rather than leveraging saliency information to address critical limitations in foreground-background disambiguation and spatial localization inherent in VLMs.

## 2.2 Foreground-background Disentanglement

Recent works identified *Foreground Bias* (Li et al., 2024b) as a fundamental limitation in VLMs, where pre-training on object-centric image-text pairs introduces systematic biases toward salient regions while neglecting holistic scene understanding. To address this issue, researchers explored foreground-background decomposition strategies for ensemble modeling of background regions.

One research direction focuses on decomposing visual embeddings into foreground and background regions at the token level. Panoptic SegFormer (Li et al., 2022b) presents a query decoupling strategy to adaptively separate visual tokens into thing and stuff queries. OpenSeeD (Zhang et al., 2023) employs language guidance to select foreground queries, which subsequently interact with learnable background queries through decoupled cross-attention blocks in the decoder. FOUND (Siméoni et al., 2023) extracts high-confident "seed" tokens to generate coarse background masks through attention maps, enhancing fine-grained object localization. ClearCLIP (Lan et al., 2024) decomposes CLIP vision encoder outputs to attention output and residual connections, learning more robust object recognition. FOCUS (You et al., 2025) leverages two pre-defined prompts to generate foreground and background masks and further calculates the contrastive loss for improved foreground localization.

Alternative approaches perform disentanglement at the class level, separating all classes into foreground and background taxonomies. DenseVLM (Li et al., 2024b) designs a novel VLM that mitigates background imbalance by generating pseudo labels for unlabeled regions using frozen VLM and then applying separate alignment objectives for pre-defined foreground and background categories. Talk2DINO (Barsellotti et al., 2024) designs a Background Cleaning Procedure that re-weights class scores based on the self-attention maps, highlighting the foreground regions while suppressing background interference. LBP (Li et al., 2024a) enhances background understanding for open-vocabulary object detection by learning background prompts from other images, effectively incorporating implicit background knowledge and achieving superior performance.

Despite these advances, existing disentanglement approaches suffer from several limitations. Token-level disentanglement fails to preserve intra-class relationships that are crucial for dense predictions. Class-level disentanglement assumes rigid foreground-background taxonomies, ignoring that instances of each class may appear as either foreground or background depending on scene composition. Furthermore, learnable disentanglement modules without any prior knowledge lack explicit guidance and are sub-optimal. In contrast to these approaches, our method leverages explicit saliency supervision to perform adaptive, class-specific foreground-background disentanglement, developing semantic coherence among ensemble representations while addressing the *Foreground Bias* in VLMs.

## 3 Methodology

### 3.1 Motivation

Our approach is motivated by the observation that the challenges of open-vocabulary semantic segmentation lie in the inherent asymmetry between foreground objects (salient, instance-centric

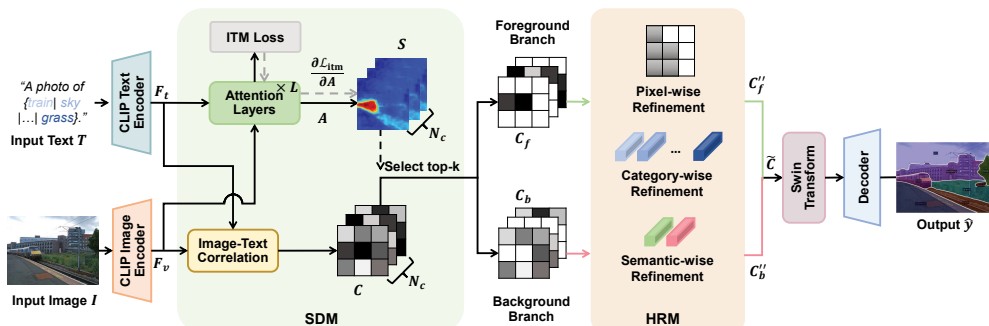

Figure 2: **Overview of the DiSa Framework.** DiSa consists of a Saliency-aware Disentanglement Module (SDM) and a Hierarchical Refinement Module (HRM), followed by an upsampling decoder.

elements) and background regions (contextual, peripheral environments), which are often entangled within shared feature spaces. To be more specific, in real-world images, the role of a category can vary based on scene composition (e.g., a train in focus vs. a train in the far background). This aligns with how humans parse visual scenes: we don't always treat "train" in different scenes with equal attention — it depends on salience, size, occlusion, etc.

While existing methods treat all regions uniformly, we argue that the inherent differences between foreground and background semantics reveal the benefits of a more principled decomposition strategy. This allows both foreground and background features to learn specialized representations and learning objectives, aligning with the "Seek common ground while reserving differences" design ideology (Li et al., 2022b). Even for *stuff* classes, the foreground refers to the most semantically informative or attribute-rich sub-regions. This behavior aligns with our above motivation: even within the same semantic category, different regions may contribute differently to the textual concept. For example, in classes like wall or sky, the textured parts of a wall and the cloud structures in the sky provide relatively stronger visual cues, while others serve as contextual or peripheral background.

To this end, we propose a divide-and-conquer formulation that leverages saliency cues to structurally decompose the segmentation task into two complementary sub-problems: foreground object localization and background region understanding. This separation not only improves robustness to dynamic scene compositions but also enhances holistic scene understanding by improving disambiguation.

## 3.2 ARCHITECTURE OVERVIEW

Fig. 2 provides an overview of our proposed framework, DiSa. Our model consists of 4 core components: a CLIP image encoder, a CLIP text encoder, a Saliency-aware Disentanglement Module (SDM) for foreground-background disentanglement, and a Hierarchical Refinement Module (HRM) for integrating multi-level fine-grained details. We follow existing works (Xian et al., 2019; Bucher et al., 2019) for the task design, e.g., the input and output formats. Given an input image $I$ and a set of text labels $T = \{T_i, i = 1, 2, ..., N_C\}$, where $N_C$ is the number of all $C$ classes, we utilize CLIP as vision-language encoders to extract image $F_v \in \mathbb{R}^{H \times W \times D}$ and text embeddings $F_t \in \mathbb{R}^{N_C \times D}$, where $D$ is the dimension size.

Our proposed pipeline begins by processing image $F_v$ and text embeddings $F_t$ to extract cross-attention maps $A \in \mathbb{R}^{HW \times N_C}$ through SDM. These attention maps are then sharpened by Image-Text Matching (ITM) loss (Li et al., 2021) gradients and the outputs are saliency maps $S_{1:N_C} \in \mathbb{R}^{H \times W \times N_C}$. Meanwhile, DiSa generates correlation maps $C_{1:N_C} \in \mathbb{R}^{H \times W \times N_C \times D}$ between image $F_v$ and text embeddings $F_t$ through cosine similarity and projection layers. All correlation tokens from $C$ are then divided into a Foreground and a Background Branch based on their corresponding saliency scores $S$. The details of SDM are explained in Section 3.3. Subsequently, we propose a three-stage Hierarchical Refinement Module (HRM) to further enhance the fine-grained localization and semantic precision of disentangled correlation maps $C_f$ and $C_b$ separately via Pixel-wise, Category-wise, and Semantic-wise Refinement. Detailed explanations of HRM are in Section 3.4. Afterwards, the refined features ($C''_f$ and $C''_b$) from the foreground and background branches are integrated through a weighted feature aggregation block to produce aggregated correlation maps $\widetilde{C} \in \mathbb{R}^{H \times W \times N_C \times D}$. Finally, it produces the final mask predictions $\hat{y}$ through an upsampling decoder.

### 3.3 SALIENCY-AWARE DISENTANGLEMENT MODULE

We design the SDM to address the inherent *Foreground Bias* in VLMs. It uses GradCAM (Selvaraju et al., 2017) to generate per-class saliency maps from cross-attention. Traditional saliency-based methods merely focus on improving model efficiency through token pruning. However, we incorporate saliency as an additional visual cue to perform complementary feature learning. The saliency is consistent with prior work that reflects semantic contribution but with a slight modification: we obtain saliency maps for each class instead of a single saliency map for all classes. In our method, saliency therefore represents regions of semantic details and informative structures, not merely regions of model confidence. Saliency provides a scalar importance score for disentanglement. It captures where meaningful evidence appears, while correlation encodes the semantic cues present in that region. The gradient-based saliency generation is essential for robust disentanglement.

**Saliency Map Generation.** Given image $F_v$ and text embeddings $F_t$, we employ cross-attention layers where text embeddings serve as the query and image embeddings serve as the key/value. The intermediate attention map $A$ captures image-text correspondences; however, the attentions typically exhibit scattered and spatially diffuse activations due to their overly broad receptive fields, which are inherent in global modeling (Wang et al., 2025). To address this limitation, we selectively suppress less-relevant regions within the attention map $A$ and enhance its spatial precision via gradient-based reweighting. This objective is achieved through an auxiliary Image-Text Matching (ITM) loss $\mathcal{L}_{\text{itm}}$ (Li et al., 2021) that provides explicit supervision for localization. Specifically, we append an auxiliary regression head to classify whether each image-text pair is matched. The ITM loss is formulated as:

$$\mathcal{L}_{\text{itm}} = \mathbb{E}_{(v,t)\sim D}\mathcal{H}(\hat{\boldsymbol{y}}_{(v,t)}^{itm}, \boldsymbol{y}^{itm}) \tag{1}$$

where $\mathcal{H}$ is the cross-entropy loss, and the ground-truth labels $\boldsymbol{y}^{itm}$ are one-hot vectors obtained from segmentation masks. During inference, we use the regressed image-text matching scores $y^{itm}$ to generate gradients, so no ground truth or class labels are required, thereby avoiding any risk of data leakage. Afterwards, we compute the gradient of $\mathcal{L}_{\text{itm}}$ and let the attention maps narrowly focus on the most discriminative regions through GradCAM-style re-weighting to produce the saliency map $S_n$ for the n-th class:

$$S_n = \max\left(0, \frac{\partial \mathcal{L}_{\text{itm}}}{\partial A_n}\right) \otimes A_n \tag{2}$$

where $\otimes$ represents element-wise multiplication. Note that we rely on the above image-text attention maps $A$, which provide robust localization cues and are continuously optimized during training through segmentation objectives, thereby stabilizing and enhancing their quality even if the auxiliary ITM supervision is imperfect.

**Foreground/Background Token Selection.** Unlike traditional approaches that treat all visual tokens uniformly, we propose a dual-branch mechanism to disentangle all visual tokens into the Foreground and Background Branches by saliency-aware feature decomposition. It enables the model to explicitly distinguish between foreground and background regions, focusing on the distinctive characteristics of both features. This architectural decoupling directly mitigates *Foreground Bias* in VLMs caused by their object-centric focus during pre-training. It enables each branch to develop domain-specific representations.

Specifically, for the n-th class, we select the top-k visual tokens in the correlation maps $C_n$ corresponding to their saliency scores $S_n$ through a binary mask as foreground correlation maps $C_{f,n}$ while the remaining are designated as background correlation maps $C_{b,n}$. These disentangled maps are then processed through two specialized branches: a Foreground Branch that models salient foreground features, and a Background Branch that captures contextual background information.

### 3.4 HIERARCHICAL REFINEMENT MODULE

Existing SOTA methods often struggle with effectively capturing the semantic boundary details in complex scenes. To address this challenge, we perform hierarchical refinement to update the correlation maps $C$ across three distinct levels: (1) **Pixel-wise Refinement**, which focuses on achieving precise spatial localization; (2) **Category-wise Refinement**, aimed at enhancing channel-wise coherence for each class; and (3) **Semantic-wise Refinement**, which captures semantic consistency within broader foreground/background groupings. The hierarchical design preserves fine-grained

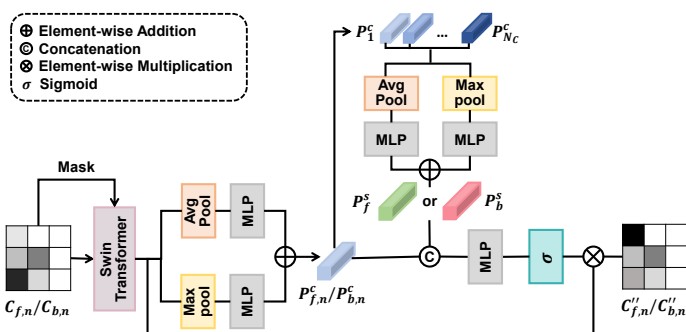

(a) **Pixel-wise Refinement**   (b) **Category-** and **Semantic-wise Refinement**

Figure 3: **Illustration of Hierarchical Refinement Module (HRM).**

spatial details while capturing cross-channel context, leading to improved segmentation precision in complex visual environments.

**Pixel-wise Refinement.** As shown in Fig. 3, Pixel-wise Refinement takes foreground correlation maps $C_f$ and background correlation maps $C_b$ as input. The Pixel-wise Refinement block is applied for spatial aggregation based on the Swin Transformer (Liu et al., 2021) block with key modifications. Instead of performing cross-attention across all tokens, we utilize masked embeddings in two branches to focus solely on foreground and background embeddings, respectively. It consists of 2 blocks: the first block implements self-attention within a local window, while the second block employs shifted window-based self-attention to enhance global context integration. The outputs are well-refined correlation maps ($C'_f$ and $C'_b$) after pixel-wise spatial aggregation, effectively suppressing noise in image-text correlations.

**Category-wise Refinement.** Subsequent to Pixel-wise Refinement, Category-wise Refinement is applied to consider category-specific cross-covariance across feature channels. Given the pixel-refined correlation maps $C'_f$ and $C'_b$, we apply 2D global average pooling and max pooling in parallel to extract both the spatial extent of target objects and discriminative clues across channel dimensions. The resulting pooled features are independently processed by 2 MLPs, and their outputs are combined through element-wise addition to generate the category prototype $P^c \in \mathbb{R}^{H \times W \times N_C \times D}$. Note that the $P^c$ is class-specific, and $P^c_{f,n}$ and $P^c_{b,n}$ represent the category prototype for the Foreground and Background Branches of the n-th class.

**Semantic-wise Refinement.** While Category-wise Refinement enhances channel-wise coherence, it is class-specific and may overlook broader contextual cues, e.g., surrounding environments or the overall scene among all classes. To improve semantic understanding of the relationship among all salient objects for the Foreground Branch and all semantic regions in the environments for the Background Branch, we design the Semantic-wise Refinement. This additional refinement block considers coarse-grained scene context across all classes, leading to more robust and generalized representations.

Similar to Category-wise Refinement, we apply 1D global average pooling and max pooling layers among category prototypes of all classes $\{P^c_i, i = 1, 2, ..., N_c\}$ to extract semantic prototypes $P^s$. Note that $P^s_f$ and $P^s_b$ are class-agnostic and shared among all classes within the Foreground and Background Branches. After extracting both category $P^c$ and semantic prototypes $P^s$, we update the pixel-refined correlation maps $C'_f$ and $C'_b$ by aggregating these channel-wise cues. Specifically, for the n-th class, we fuse both $P^c_n$ and $P^s$ by concatenation. Note that the semantic prototype used is selected based on the branch: $P^s_f$ for the Foreground Branch and $P^s_b$ for the Background Branch. This fused output is then element-wise multiplied by the pixel-refined correlation embeddings $C'$, followed by a sigmoid activation as follows:

$$C''_{f,i} = C'_{f,i} \otimes \sigma(\text{MLP}(\text{Concate}(P^c_{f,i} + P^s_f))) \qquad (3)$$

$$C''_{b,i} = C'_{b,i} \otimes \sigma(\text{MLP}(\text{Concate}(P^c_{b,i} + P^s_b))) \qquad (4)$$

where $i = 1, 2, ..., N_C$, Concate is concatenation, $\sigma(\cdot)$ denotes sigmoid function, and $\otimes$ refers to element-wise multiplication.

| Model | VLM | Additional Backbone | Training Dataset | Additional Dataset | A-847 | PC-459 | A-150 | PC-59 | PAS-20 | PAS-20$^b$ |
|---|---|---|---|---|---|---|---|---|---|---|
| LSeg [arXiv21] | CLIP ViT-B/32 | ResNet-101 | PASCAL VOC-15 | ✗ | - | - | - | - | 47.4 | - |
| LSeg+ [ECCV22] | ALIGN | ResNet-101 | COCO-Stuff | ✗ | 2.5 | 5.2 | 13.0 | 36 | - | 59.0 |
| ZegFormer [CVPR22] | CLIP ViT-B/16 | ResNet-101 | COCO-Stuff-156 | ✗ | 4.9 | 9.1 | 16.9 | 42.8 | 86.2 | 62.7 |
| ZegFormer [CVPR22] | CLIP ViT-B/16 | ResNet-101 | COCO-Stuff | ✗ | 5.6 | 10.4 | 18.0 | 45.5 | 89.5 | 65.5 |
| ZSseg [ECCV22] | CLIP ViT-B/16 | ResNet-101 | COCO-Stuff | ✗ | 7.0 | - | 20.5 | 47.7 | 88.4 | - |
| OpenSeg [ECCV22] | ALIGN | ResNet-101 | COCO Panoptic | ✓ | 4.4 | 7.9 | 17.5 | 40.1 | - | 63.8 |
| OVSeg [CVPR23] | CLIP ViT-B/16 | ResNet-101 | COCO-Stuff | ✓ | 7.1 | 11.0 | 24.8 | 53.3 | 92.6 | - |
| ZegCLIP [CVPR23] | CLIP ViT-B/16 | - | COCO-Stuff-156 | ✗ | - | - | - | 41.2 | 93.6 | - |
| SAN [CVPR23] | CLIP ViT-B/16 | - | COCO-Stuff | ✗ | 10.1 | 12.6 | 27.5 | 53.8 | 94.0 | - |
| EBSeg [CVPR24] | CLIP ViT-B/16 | - | COCO-Stuff | ✗ | 11.1 | 17.3 | 30.0 | 56.7 | 94.6 | - |
| SED [CVPR24] | ConvNeXt-B | - | COCO-Stuff | ✗ | 11.4 | 18.6 | 31.6 | 57.3 | 94.4 | - |
| CAT-Seg [CVPR24] | CLIP ViT-B/16 | - | COCO-Stuff | ✗ | _12.0_ | 19.0 | 31.8 | 57.5 | 94.6 | _77.3_ |
| DPSeg [CVPR25] | CLIP ViT-B/16 | - | COCO-Stuff | ✗ | _12.0_ | _19.5_ | _32.9_ | _58.1_ | _96.0_ | - |
| **DiSa** | CLIP ViT-B/16 | - | COCO-Stuff | ✗ | **12.6** | **20.3** | **33.7** | **59.3** | **97.0** | **79.9** |
|  |  |  |  |  | (+0.6) | (+0.8) | (+0.8) | (+1.2) | (+1.0) | (+2.6) |
| LSeg [arXiv21] | CLIP ViT-B/32 | ViT-L/16 | PASCAL VOC-15 | ✗ | - | - | - | - | 52.3 | - |
| OpenSeg [ECCV22] | ALIGN | Eff-B7 | COCO Panoptic | ✓ | 8.1 | 11.5 | 26.4 | 44.8 | - | 70.2 |
| OVSeg [CVPR23] | CLIP ViT-L/14 | Swin-B | COCO-Stuff | ✓ | 9.0 | 12.4 | 29.6 | 55.7 | 94.5 | - |
| SAN [CVPR23] | CLIP ViT-L/14 | - | COCO-Stuff | ✗ | 12.4 | 15.7 | 32.1 | 57.7 | 94.6 | - |
| ODISE [CVPR23] | CLIP ViT-L/14 | Stable Diffusion | COCO-Stuff | ✗ | 11.1 | 14.5 | 29.9 | 57.3 | - | - |
| SCAN [CVPR24] | CLIP ViT-L/14 | - | COCO-Stuff | ✗ | 14.0 | 16.7 | 33.5 | 59.3 | 97.2 | - |
| EBSeg [CVPR24] | CLIP ViT-L/14 | - | COCO-Stuff | ✗ | 13.7 | 21.0 | 32.8 | 60.2 | 96.4 | - |
| SED [CVPR24] | ConvNeXt-L | - | COCO-Stuff | ✗ | 13.9 | 22.6 | 35.2 | 60.6 | 96.1 | - |
| CAT-Seg [CVPR24] | CLIP ViT-L/14 | - | COCO-Stuff | ✗ | _16.0_ | _23.8_ | _37.9_ | _63.3_ | 97.0 | _82.5_ |
| DPSeg [CVPR25] | CLIP ViT-L/14 | - | COCO-Stuff | ✗ | 14.9 | 23.5 | 36.4 | 62.0 | _97.4_ | - |
| **DiSa** | CLIP ViT-L/14 | - | COCO-Stuff | ✗ | **16.3** | **24.9** | **38.9** | **64.7** | **98.7** | **84.7** |
|  |  |  |  |  | (+0.3) | (+1.1) | (+1.0) | (+1.4) | (+1.3) | (+2.2) |

Table 1: **Quantitative results on 6 benchmarks.** The best-performing results are presented in **bold**, while the second-best results are underlined. Improvements over the second-best are in **bold**.

By hierarchically refining correlation maps across pixel-, category-, and semantic-levels, HRM makes the saliency-aware correlation representations more informative and enhances the accuracy, leading to fine-grained spatial precision in downstream segmentation tasks.

### 3.5 Foreground and Background Aggregation

To uniformly model all pixels after capturing foreground- and background-specific features, we aggregate the disentangled refined correlation maps ($C''_{f,n}$ and $C''_{b,n}$) for the n-th class using the previous binary mask. We further employ a Swin Transformer block to mitigate potential misalignment between dual branches, and the aggregated correlation maps are denoted as $\widetilde{C}$. Finally, $\widetilde{C}$ serves as visual guidance and are fed into an upsampling decoder, along with image embeddings $F_v$ from the CLIP image encoder, to generate the final mask predictions $\hat{y}$.

## 4 Experiments

### 4.1 Datasets

Our experiments are trained on COCO-Stuff (Caesar et al., 2018) and evaluated on 6 large-scale semantic segmentation datasets . **ADE20K** (Zhou et al., 2019) is a large-scale benchmark for semantic segmentation with 2000 validation images, supporting two evaluation protocols: ADE-150 with 150 categories, and ADE-847 with extended 847 classes. **PASCAL-VOC** (Everingham et al., 2010) is a widely used dataset containing 1,500 validation images with 20 foreground categories, referred as PAS-20. Another evaluation protocol PAS-20$^b$ (Ghiasi et al., 2022) with one extra class for background is also included. **PASCAL-Context** (Mottaghi et al., 2014) extends PASCAL VOC, supporting 2 evaluation protocols: PC-59 with 59 labeled classes and PC-459 with 459 categories.

### 4.2 Implementation Details

We implement our work using PyTorch (Paszke, 2019) and Detectron2 (Wu et al., 2019). The loss function is a weighted sum of cross-entropy loss and the ITM loss. We set $D = 128$, and downsample the feature maps to $\frac{H}{4} \times \frac{W}{4}$ resolution. The $k$ parameter for selecting foreground tokens is 96. The decoder consists of 2 transposed convolution layers that take $\widetilde{C}$ and $F_v$ as inputs. Following CAT-Seg (Cho et al., 2024), we fine-tune query and key projections in attention layers of CLIP image and text encoders. We train the model using the AdamW optimizer (Loshchilov & Hutter, 2017) with batch size 2. The learning rate is 2e-4 for our designed modules and 2e-6 for CLIP encoders. We use 2 NVIDIA RTX A5000 GPUs for training. All of the models are trained for 80,000 iterations.

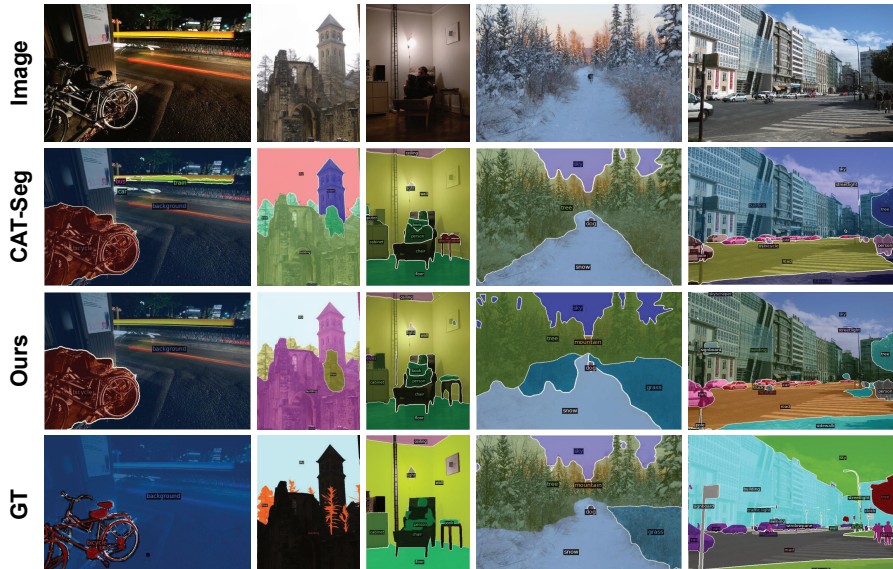

Figure 4: **Qualitative results compared to CAT-Seg.** DiSa produces more accurate predictions of small objects and visually-similar regions compared to existing SOTA methods. More qualitative results are in the Appendix.

## 4.3 QUANTITATIVE RESULTS

Table 1 demonstrates quantitative results on standard open-vocabulary semantic segmentation datasets (Zhou et al., 2019; Everingham et al., 2010; Mottaghi et al., 2014). We compare existing works, LSeg (Li et al., 2022a), LSeg+ (Ghiasi et al., 2022), ZegFormer (Ding et al., 2022), ZSseg (Xu et al., 2022), OpenSeg (Ghiasi et al., 2022), OVSeg (Liang et al., 2023), ZegCLIP (Zhou et al., 2023), SAN (Xu et al., 2023b), ODISE (Xu et al., 2023a), SCAN (Liu et al., 2024), EBSeg (Shan et al., 2024), SED (Xie et al., 2024), CAT-Seg (Cho et al., 2024), and DPSeg (Zhao et al., 2025) with similar-scale VLMs. Note that we adopt the DPSeg (Zhao et al., 2025) inference I model for fair comparison. Unlike some prior works, our model does not leverage any additional datasets or backbones.

Our method, DiSa, achieves consistent and significant gains across all benchmarks, in both base-VLM and large-VLM settings. As shown in Table 1, in the base-VLM configuration, DiSa outperforms prior SOTA approaches with the improvements of +0.6%, +0.8%, +0.8%, +1.2%, +1.0%, and +2.6% mIoU (with an average performance gain of +1.2% mIoU), on A-847, PC-459, A-150, PC-59, PAS-20, and PAS-20$^b$, respectively. For the large-VLM configuration, DiSa outperforms DPSeg by +0.3%, +1.1%, +1.0%, +1.4%, +1.3%, and +2.2% mIoU with an average gain of 1.2% mIoU among all datasets. Note that DiSa has the most significant relative performance gains on PAS-20$^b$, demonstrating that it effectively mitigates **Foreground Bias** in VLMs. These gains are not only statistically meaningful but also practically significant given the performance saturation observed in open-vocabulary segmentation tasks. Model efficiency analysis is in the Appendix.

We attribute the leading performance of DiSa to two factors: (1) Our proposed Saliency-aware Disentanglement enhances context-aware features while preserving semantic coherence. It effectively mitigates **Foreground Bias**, as demonstrated by significant improvements on PAS-20$^b$, which includes background classes. (2) Hierarchical Refinement Module yields accurate and robust boundaries via multi-level refinement, contributing to consistent performance gains across all datasets.

## 4.4 QUALITATIVE RESULTS

We evaluate qualitative results of our method with CAT-Seg (Cho et al., 2024) using default settings in Fig. 4. We present diverse scenarios, including crowded background (columns 1&5) and visually similar classes (rows 2-4). CAT-Seg struggles to handle complex foreground-background relations and locate accurate boundaries. For example, in column 1, the background is misclassified as "train". Similarly, in columns 2&4, CAT-Seg produces ambiguous boundaries between visually similar categories (e.g., "snow" and "grass"), reflecting its limited capacity for fine-grained spatial

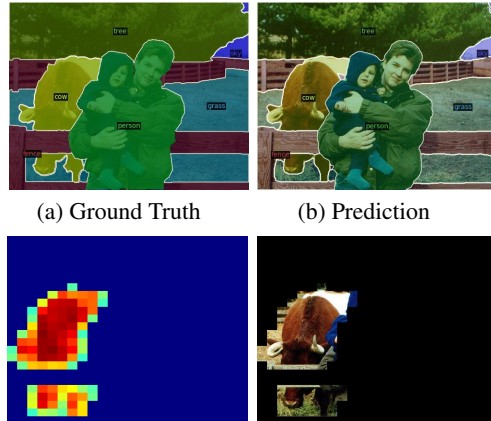

(a) Ground Truth   (b) Prediction

(c) Foreground saliency   (d) Foreground tokens

Figure 5: **Qualitative comparison between saliency and predictions.**

| Model | # Params. (M) | GFLOPs | Inference time (s) |
|---|---|---|---|
| ZegFormer | 531.2 | 19,425.6 | 3.10 |
| ZSseg | 530.8 | 22,302.1 | 3.11 |
| OVSeg | 532.6 | 19,345.6 | 2.98 |
| CAT-Seg | 433.7 | 2,121.1 | 0.78 |
| ESC-Net | 451.3 | 2,203.5 | 0.76 |
| Ours | 456.2 | 2,287.3 | 0.69 |

Table 2: **Model complexity comparison.** We use CLIP ViT-B/16 for VLM and one single A6000 GPU for fair comparison.

| Decomposition | A-847 | PC-459 | A-150 | PC-59 | PAS-20 | PAS-20[b] |
|---|---|---|---|---|---|---|
| (I) DiSa w/o SDM | 11.4 | 18.8 | 30.9 | 56.3 | 94.5 | 76.1 |
| (II) Token-level | 12.1 | 19.9 | 32.6 | 58.2 | 94.5 | 78.9 |
| (III) Class-level | 11.5 | 19.5 | 31.1 | 58.2 | 95.1 | 78.5 |
| (IV) DiSa w/o HRM | 11.7 | 18.7 | 31.2 | 57.7 | 95.3 | 78.5 |
| (V) (IV) + Pixel | 12.4 | 19.5 | 32.3 | 58.8 | 95.9 | 78.9 |
| (VI) (IV) + Category | 11.8 | 19.1 | 32.0 | 57.9 | 95.4 | 78.3 |
| (VII) (V) + Category | 12.5 | 19.8 | 33.1 | 59.0 | 96.6 | 79.1 |
| (VIII) **Ours** | **12.6** | **20.3** | **33.7** | **59.3** | **97.0** | **79.9** |

Table 3: **Ablation study for various design choices.** CLIP ViT-B/16 is used as VLM for ablation.

localization. In contrast, DiSa preserves object integrity in crowded scenes, demonstrating superior robustness in challenging scenarios.

We present visualizations of foreground saliency and image tokens of a specific class "cow" in Fig. 5. We observe that foreground and background tokens of the partially occluded cow are identified and not suppressed by other categories. It is consistent with our design, yielding sharper boundaries.

### 4.5 MODEL EFFICIENCY ANALYSIS

We further conduct the model efficiency analysis (parameter size and GFLOPS) on all 6 datasets in Table 2. Notably, ZegFormer (Ding et al., 2022), ZSseg (Xu et al., 2022), and OVSeg (Liang et al., 2023) rely on large-scale backbones and complex vision-language fusion modules, with more than 530M parameters and 19k GFLOPs. In contrast, our model significantly reduces inference cost to 2k GFLOPs while maintaining a competitive parameter count of 456M. Although slightly larger than CAT-Seg (Cho et al., 2024) and ESC-Net (Lee et al., 2025), our framework achieves comparable efficiency and remains lightweight compared to other CLIP-based methods. These results demonstrate that DiSa dual-branch design and hierarchical refinement modules introduce minimal overhead while delivering strong performance, highlighting its efficiency in balancing performance with computational costs.

### 4.6 ABLATION STUDY

**Design choices for disentanglement.** To validate the impact of our saliency-aware disentanglement, we compare our **(I)** baseline (DiSa without SDM) with 2 other designs in Table 3: **(II)** token-level, following PraNet (Hu et al., 2025) to decouple foreground/background features, **(III)** class-level, leveraging Large Language Models (Achiam et al., 2023) for a pre-defined taxonomy and separating all classes into 2 branches. As shown in the table, token-level disentanglement **(II)** achieves marginal improvements over the baseline (with an average gain of 0.66%). Class-level disentanglement **(III)** slightly improves on some benchmarks, likely due to its rigidity in adapting to varying scene contexts. In contrast, our proposed saliency-aware disentanglement **(VIII)** consistently outperforms other decompositions by 1.1% (token-level) and 1.5% (class-level) on average, demonstrating its efficacy.

Notably, it yields a substantial improvement (2.6%) on PAS-20$^b$, effectively alleviating ***Foreground Bias***.

**Component analysis for HRM.** To validate the effectiveness of HRM, we further evaluate the performance gain of four variants **(IV-VII)** by gradually adding their components to the baseline in Table 3. Specifically, they are: **(IV)** baseline (DiSa without HRM), **(V)** adding Pixel-wise Refinement to **(IV)**, **(VI)** adding Category-wise Refinement to **(IV)**, **(VII)** adding Category-wise Refinement to **(V)**, and **(VIII)** employing all designed components. Introducing Pixel-wise Refinement **(V)** improves the average mIoU by 0.78%. Adding Category-wise Refinement **(VII)** further boosts performance by capturing channel-wise category semantics, with an average gain of 1% over the baseline. Finally, incorporating Semantic-wise Refinement **(VIII)** yields the highest overall performance (1.62% on average). It demonstrates that HRM and multi-level refinement are essential for mitigating ***Limited Spatial Localization*** and semantic coherence.

## 5 CONCLUSION

In this paper, we propose DiSa, a novel Saliency-aware Foreground-background Disentangled framework for open-vocabulary semantic segmentation. To address the ***Foreground Bias*** and ***Limited Spatial Localization*** limitations in VLMs, we propose a Saliency-aware Disentanglement Module (SDM), which performs adaptive foreground-background decomposition based on saliency cues, enabling context-dependent ensemble feature learning. Additionally, by integrating a Hierarchical Refinement Module (HRM), DiSa yields fine-grained spatial localization through Pixel-, Category-, and Semantic-wise Refinement. Extensive experimental evaluations on six large-scale datasets demonstrate the effectiveness of our model. Our observations and novel design shift the paradigm and suggest a promising direction for future research.

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

APPENDIX OVERVIEW

In this Appendix, we provide additional details of the paper, including other model details (Section A), other implementation details (Section B), additional ablation study (Section C), additional qualitative results (Section D), and limitations of proposed method (Section F).

## A  OTHER MODEL DETAILS

### A.1  UPSAMPLING DECODER

We adopt a lightweight upsampling decoder following the design in (Cho et al., 2024). Specifically, we extract intermediate visual embeddings from the 4-th and 8-th layers (Dosovitskiy et al., 2020) of the CLIP ViT-L/14 image encoder (or the 8-th and 16-th layers of CLIP ViT-L/14) for higher-resolution guidance. The decoder consists of two identical transposed convolution layers that progressively upsample the feature maps. It takes the correlation maps with a resolution of $24 \times 24$ as input, and outputs predictions at a resolution of $96 \times 96$.

The effectiveness of this simple decoder stems from our saliency-aware disentanglement and hierarchical refinement, which models rich contexts while preserving accurate object boundaries, thereby enhancing feature extraction qualities.

## B  OTHER IMPLEMENTATION DETAILS

**Training details.** We use pre-trained CLIP ViT-B/16 (Radford et al., 2021) as our base-VLM and CLIP ViT-L/14 (Radford et al., 2021) as our large-VLM, following the same setting as in most of the recent SOTA models (Cho et al., 2024). We train three attention layers for generating saliency maps following the empirical experience and ablation studies in PnP-OVSS (Luo et al., 2024). For the Pixel-Wise Refinement and subsequent Swin Transformer (Liu et al., 2021) used for aggregation, we adopt the commonly used structure: one non-shifted window attention layer, followed by a shifted window attention layer. Some other training details include Warmup Cosine Learning Rate scheduler (Gotmare et al., 2018) and 1e-4 weight decay.

**Data preprocessing.** The data augmentation used in our work includes random cropping, and photometric distortion, following (Cheng et al., 2022). During training, saturation, hue, and contrast are randomly adjusted for robustness. The training resolution is set to be $384 \times 384$.

**Text template.** We utilize the commonly used prompt template for text labels, which is "A photo of a class", without relying on cutting-edge templates. We do not incorporate any LLM-generated or handcrafted prompts in our work.

**Evaluation metrics.** We use mean Intersection over Union (mIoU) to measure segmentation performance. For model efficiency analysis, we use parameter size and GFLOPs.

## C  ADDITIONAL ABLATION STUDY

**Design choices of foreground selection $k$.** To further validate the impact of foreground selection hyperparameter, we leverage different $k$ to evaluate our model's robustness in Table 4: **(I)** $k$=16 (3% ratio of all tokens), **(II)** $k$=48 (8% ratio of all tokens), **(III)** $k$=96 (17% ratio of all tokens). Overall, performance gradually improves with increasing $k$, and it yields small but consistent gains when $k$=96, achieving a good balance between expressiveness (for small $k$) and noise (for large $k$). PC-459 and A-847 improve the most across $k$, suggesting they benefit significantly from foreground-background disentanglement. Results on PAS-20 are relatively robust, which reveals that our disentanglement design captures complex scenario understanding and mitigates ***Foreground Bias***.

## D  ADDITIONAL QUALITATIVE RESULTS

To further validate our model, we present more visualizations of qualitative results on A-150 (Zhou et al., 2019) in Fig. 7, A-847 (Zhou et al., 2019) in Fig. 8, PC-59 (Everingham et al., 2010) in Fig. 9,

| $k$ | A-847 | PC-459 | A-150 | PC-59 | PAS-20 | PAS-20$^b$ |
|---|---|---|---|---|---|---|
| (I) 16 | 11.4 | 18.7 | 32.6 | 57.8 | 96.3 | 78.0 |
| (II) 48 | 12.0 | 19.8 | **34.1** | 58.3 | **97.1** | 78.6 |
| (IV) **96** | **12.6** | **20.3** | 33.7 | **59.3** | 97.0 | **79.9** |

Table 4: **Ablation study for foreground selection hyperparameter $k$.**

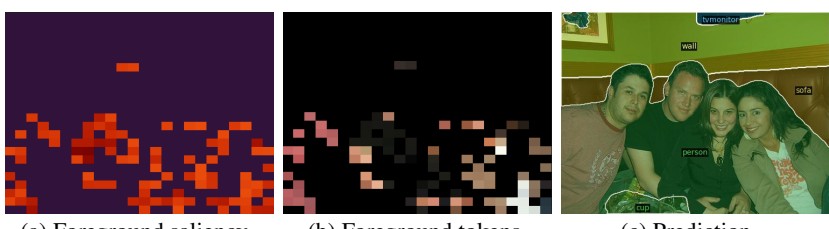

|  (a) Foreground saliency  |  (b) Foreground tokens  |  (c) Prediction  |
|---|---|---|

Figure 6: **Failure Case with imprecise foreground/background disentanglement.**

and PC-459 (Everingham et al., 2010) in Fig. 10. DiSa consistently produces accurate and robust predictions in complex scenarios, demonstrating its efficacy.

We additionally present the comparison of qualitative results on PAS-20$^b$ (Ghiasi et al., 2022) between DiSa and one previous SOTA approach, CAT-Seg (Cho et al., 2024), in Fig. 11. Note that PAS-20$^b$ has one extra "background" class, and the results clearly illustrate our identified limitations and motivations. Specifically, CAT-Seg struggles to (i) separate foreground and background areas (e.g., window in row 1, potted plant in row 2, and sofa in row 5), and (ii) define accurate boundaries between objects (e.g., TV monitor in row 3 and bicycle in row 4). These two limitations correspond to the ***Foreground Bias*** and ***Limited Spatial Localization*** inherent in VLMs, respectively. In contrast, DiSa improves foreground-background contexts and generates more precise object boundaries, demonstrating DiSa's ability to tackle challenging scenarios and mitigate the aforementioned limitations.

# E  FAILURE CASES

We additionally provide a failure case of imperfect foreground/background separation in Fig. 6. In this crowded scene, some salient regions of the class "person" are not assigned as the foreground. It demonstrates that, for objects that vary widely in size, the disentanglement might become unstable, leading to inaccurate or ambiguous foreground/background separation. However, the ensemble nature of our dual branches provides robustness by preserving complementary cues in the alternative branch compensate for such errors, leading to more reliable fused predictions.

# F  LIMITATION

Following prior state-of-the-art works (Cho et al., 2024; Tang et al., 2024), we evaluate our model on standard open-vocabulary semantic segmentation datasets such as COCO-Stuff (Caesar et al., 2018) and ADE20K (Zhou et al., 2019). However, these datasets contain incorrect or ambiguous ground-truth annotations, raising concerns about the reliability of evaluation. This highlights the need for building a new, high-quality dataset for the task.

# G  THE USE OF LARGE LANGUAGE MODELS (LLMS)

We used LLMs to improve the writings. It was used to check typos, grammar and style issues, and resolve minor notation inconsistencies. It also suggested alternative phrasings for clarity. The LLM did not contribute to research ideas and model design. All suggested edits were reviewed by the authors before incorporation.

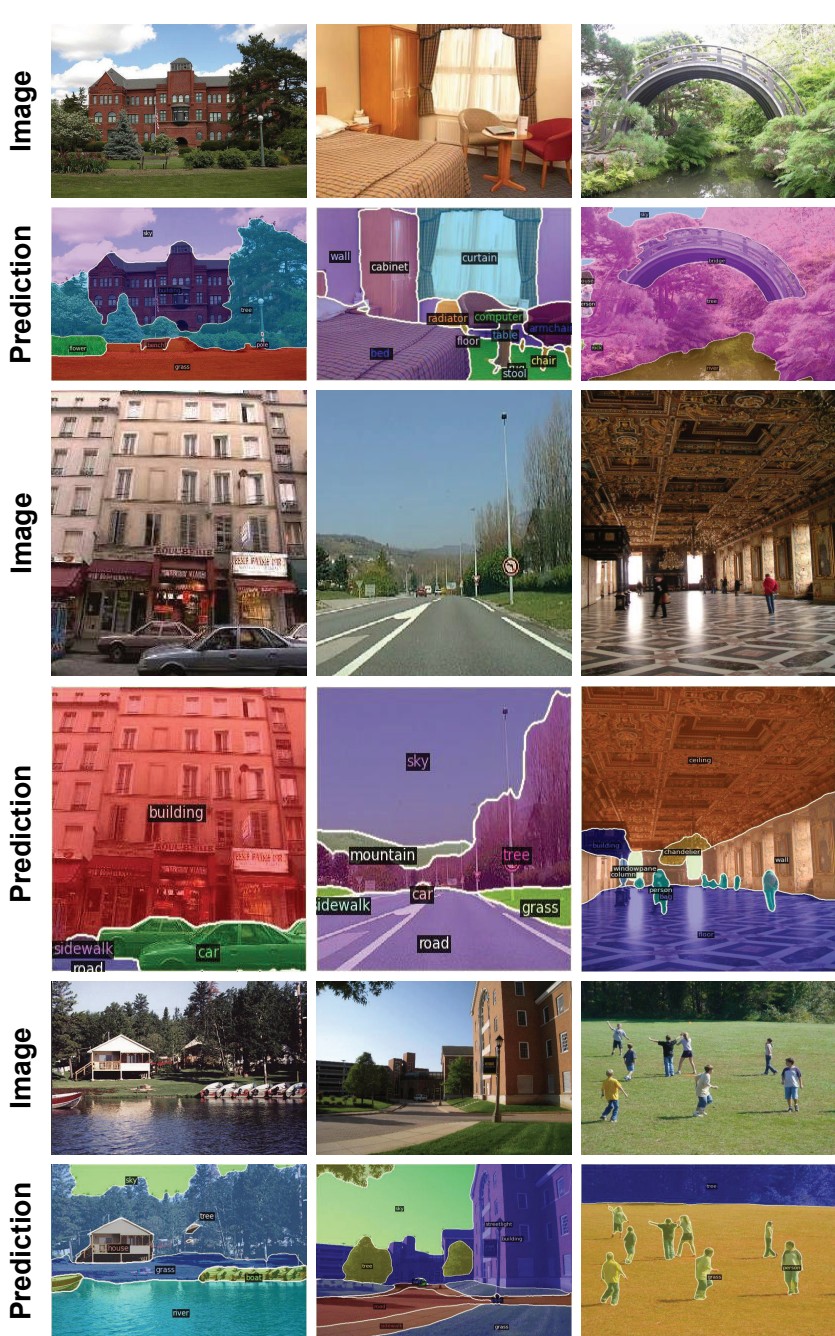

Figure 7: **Qualitative results on ADE20K with 150 classes.**

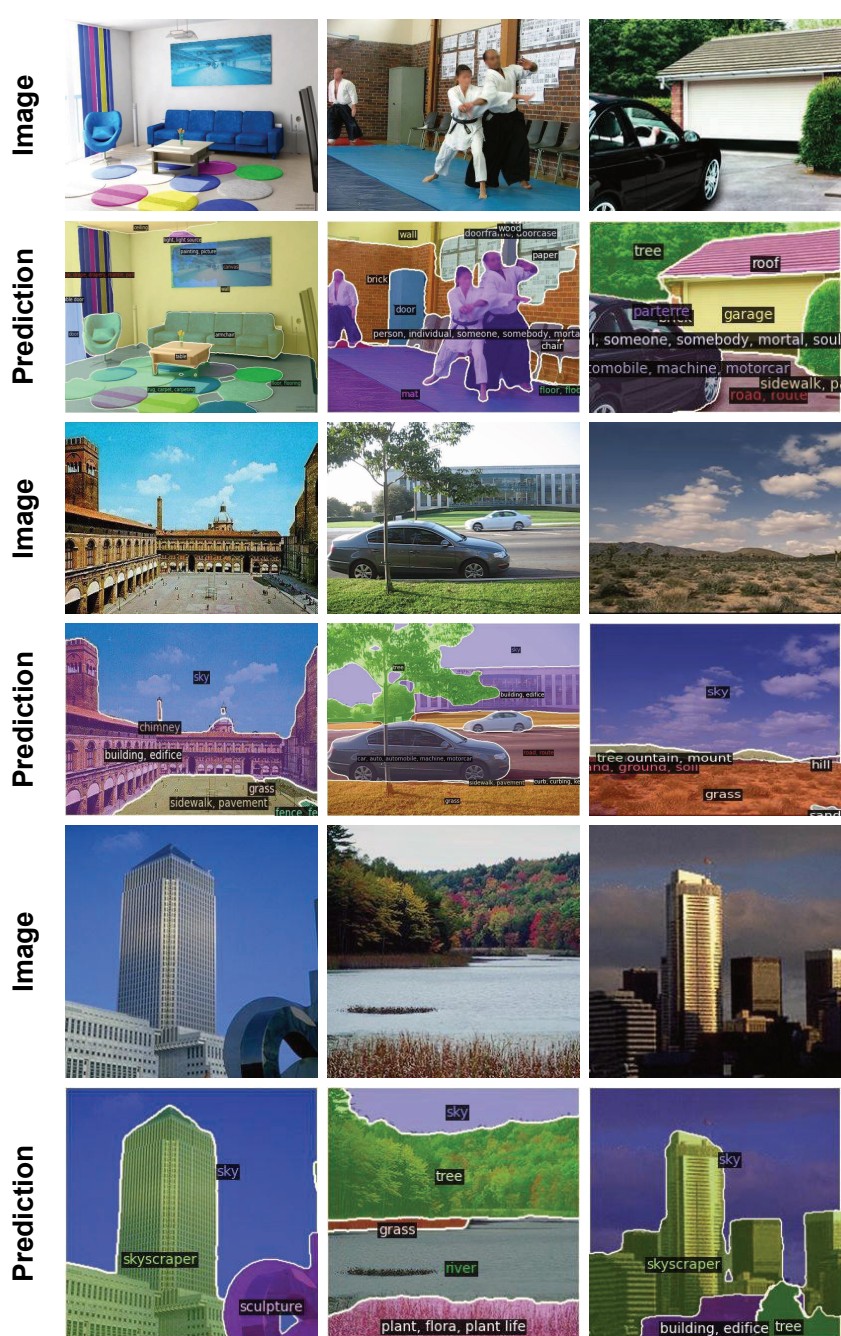

Figure 8: **Qualitative results on ADE20K with 847 classes.**

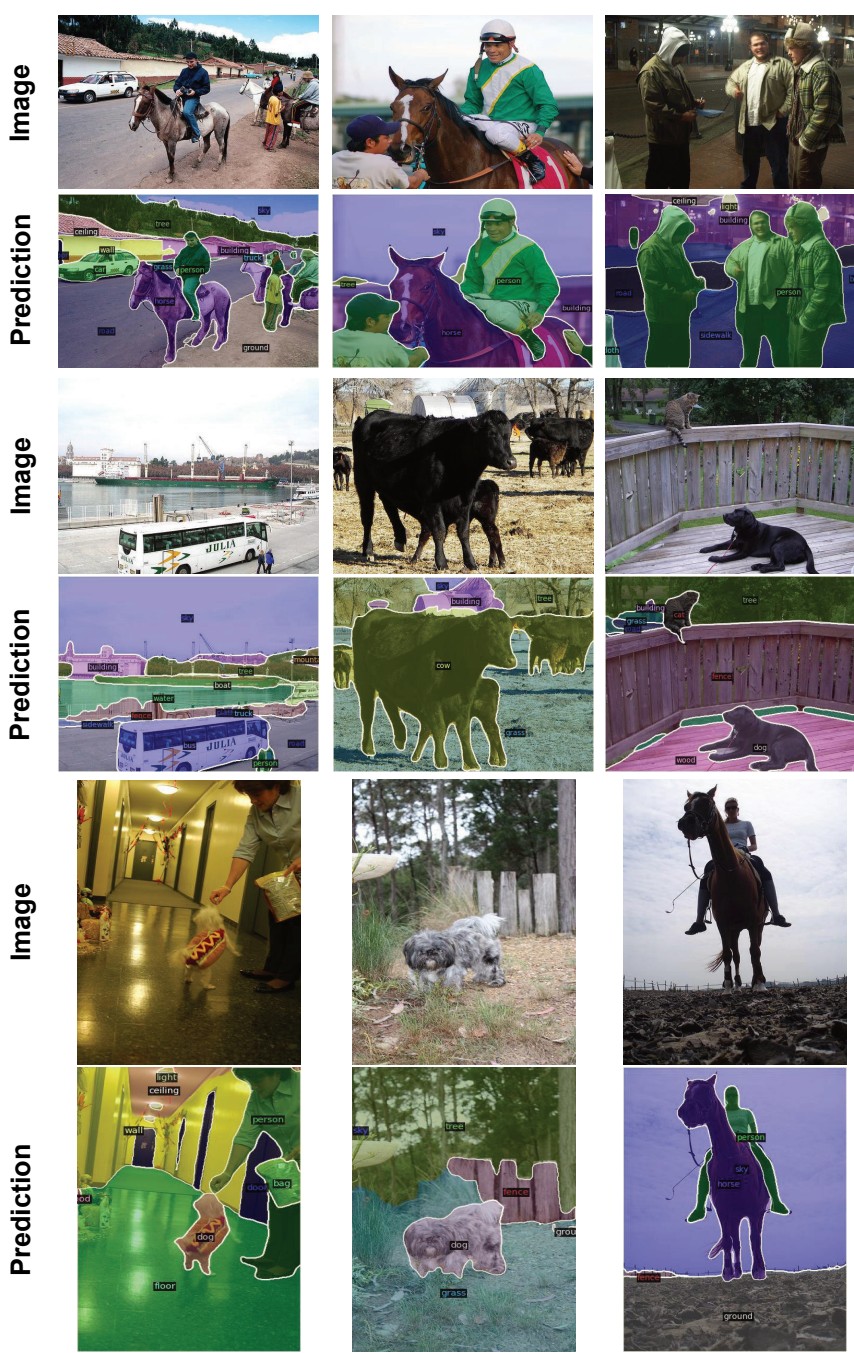

Figure 9: **Qualitative results on PASCAL Context with 59 classes.**

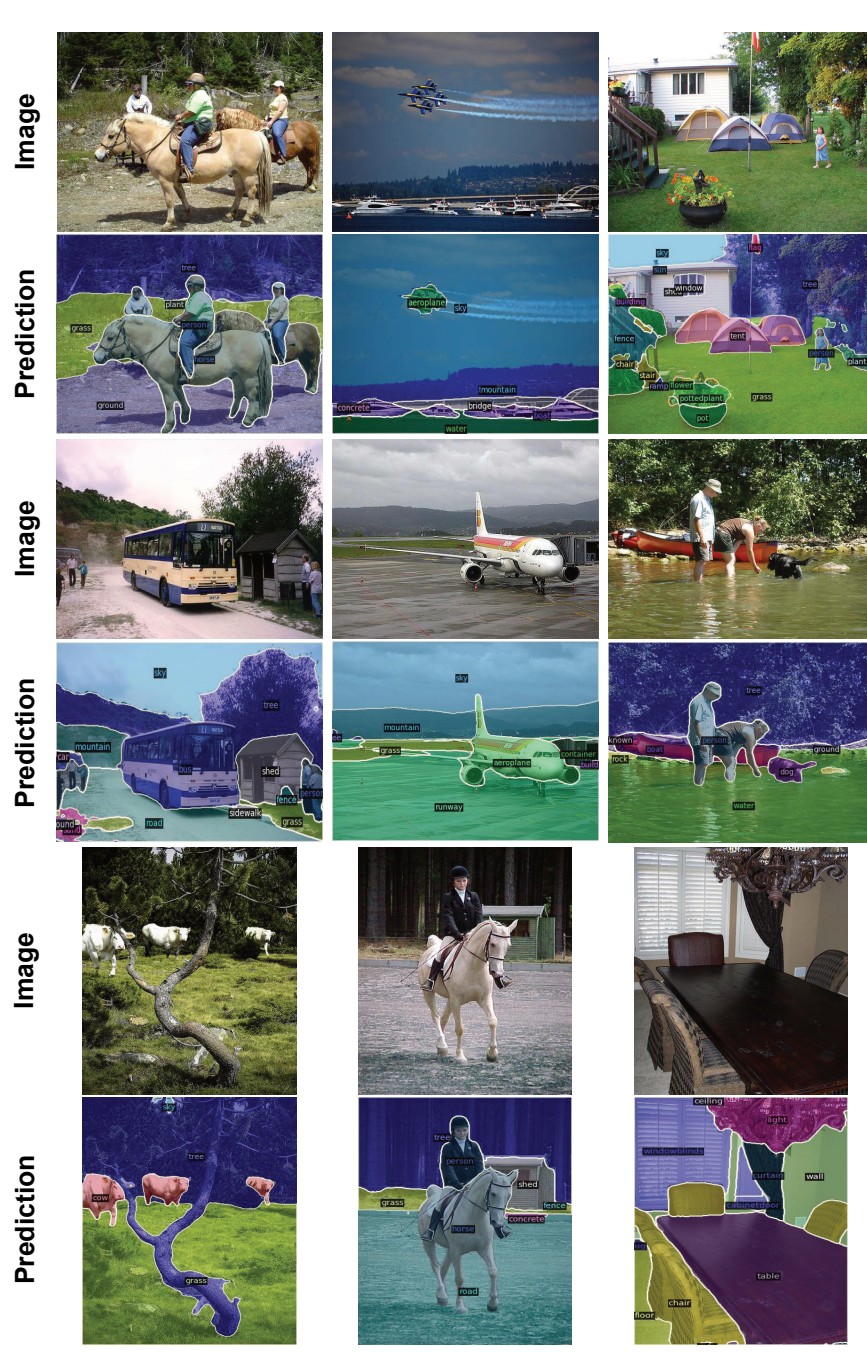

Figure 10: **Qualitative results on PASCAL Context with 459 classes.**

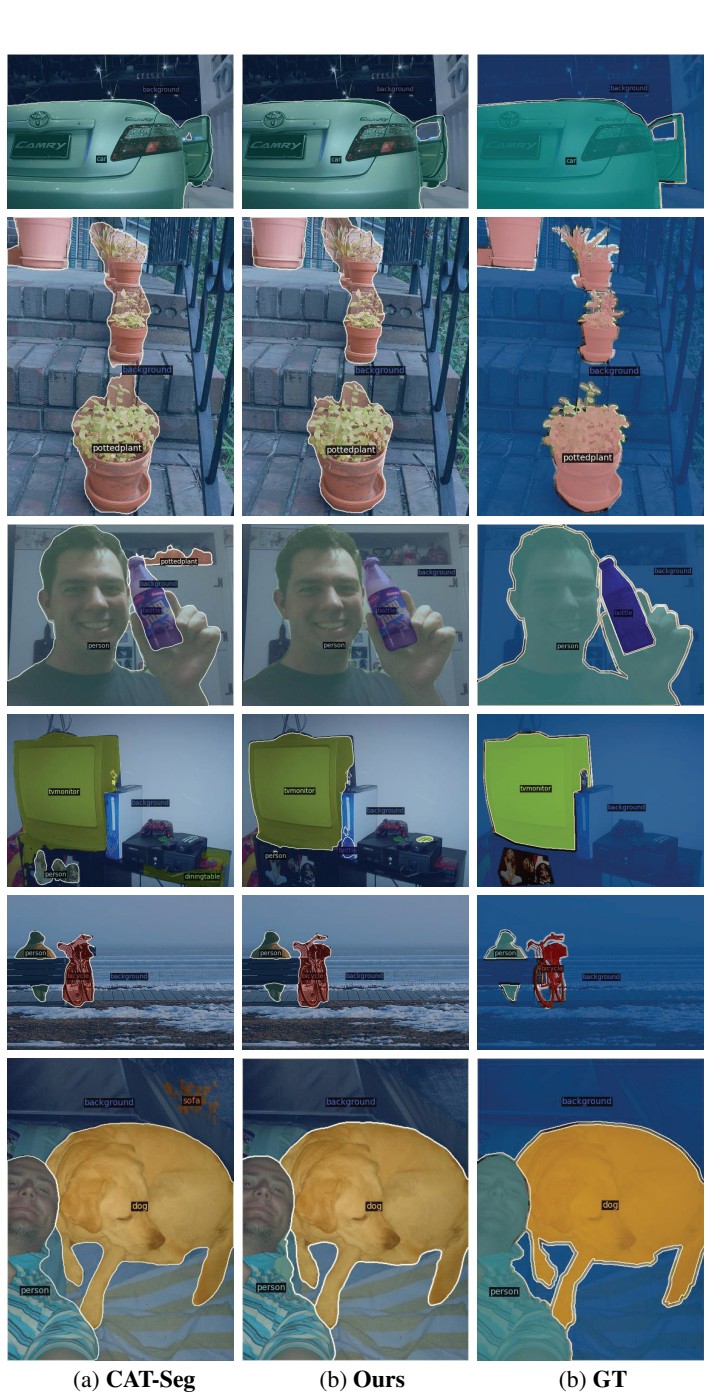

(a) **CAT-Seg**          (b) **Ours**          (b) **GT**

Figure 11: **Comparison of Qualitative results on PAS-20$^b$.** We compare DiSa with CAT-Seg.

