# OpenReview forum: "DiSa: Saliency-Aware Foreground-Background Disentangled Framework for Open-Vocabulary Semantic Segmentation"
_ICLR.cc/2026/Conference — Submitted to ICLR 2026_

### Official Review · Reviewer_RfyL · 2025-10-27

**Soundness:** 2
**Presentation:** 1
**Contribution:** 2
**Rating:** 2
**Confidence:** 3

**Summary:**

This paper introduces DiSa, a new framework for open-vocabulary semantic segmentation . The authors try to solve two key limitations in existing Vision-Language Models (VLMs) like CLIP, Foreground Bias and Limited Spatial Localization.  The paper claims that this dual-branch, saliency-guided approach allows the model to learn specialized representations that better handle both foreground objects and background context.

**Strengths:**

The paper's primary strength lies in its clear and accurate motivation. The identification of "Foreground Bias" and "Limited Spatial Localization" as two distinct, critical failures of VLMs in dense prediction tasks is sharp and well-articulated. The proposed disentanglement strategy bring additional performance improvement over baseline.

**Weaknesses:**

The submission, in its current form, suffers from several critical weaknesses that prevent me from recommending acceptance. My primary concerns are as follows:

1.  **Fundamentally Unsound Methodology and Notations:** The paper's core contribution, the Saliency-aware Disentanglement Module (SDM), is described in a manner that is mathematically incoherent and contradictory.
    * In Section 3.2, the authors define their key tensors, the saliency map $S$ and correlation map $C$, in two conflicting ways *in the same sentence*: first as a set (e.g., $S = \{S_i, ...\}$) and immediately after as a single tensor (e.g., $S \in \mathbb{R}^{H \times W \times N_C \times D}$). This is a fundamental error.
    * The specified tensor dimensions are nonsensical. A "saliency map" or a "correlation map" (derived from cosine similarity) should be scalar-valued, with dimensions $\mathbb{R}^{H \times W \times N_C}$. The authors' inclusion of an arbitrary feature dimension $D=128$ is technically incorrect and is never justified. This error makes the central mechanism of the paper—using $S$ to filter $C$—impossible to understand or reproduce.

2.  **Overly Convoluted and "Chaotic" Pipeline:** The pipeline in Fig.2 generates correlation maps $C$, while *in parallel* generating saliency maps $S$ by "sharpening" cross-attention maps $A$ with gradients from an $\mathcal{L}{itm}$. The paper provides no clear justification for why this complex, gradient-based saliency generation (using $\partial \mathcal{L}_{itm} / \partial A$) is necessary or superior to simpler, more direct methods. The design feels arbitrary and poorly motivated.

3. **Marginal Ablation Gains:** The ablation study fails to justify the added complexity. As highlighted, row (VI) (`(IV) + Category`) shows scores (11.8, 19.1, 32.0, 57.9, 95.4, 78.3) that are only marginally better, and in some cases worse, than the main (I) Baseline (12.0, 19.0, 31.8, 57.5, 94.6, 77.3). This suggests the proposed refinement modules offer minimal practical benefit for their complexity.

4.  **Formatting Inconsistencies:** Figure 6, labeled as a "Figure," is in fact a table presenting numerical data.

**Questions:**

please check weakness

---

> ### Author Response · Authors · 2025-11-20
> **Official Comment by Authors**
>
> We sincerely thank the reviewer for the constructive feedback. We greatly appreciate your recognition of the paper’s motivation and performance improvement. The comments guide us in improving the clarity and impact of our work. We address each concern below:
>
> > Unsound notations.
>
> We thank the reviewer for raising this important point and we respectfully disagree with it. We clarify that:
>
> **(a) Clarification of notation $S$ is intentional and not contradictory**
>
> The notation $S=\\{S_i\\}$ is used to indicate that **we obtain one saliency map per class**, where each $S_i$ corresponds to class $i$. In the same expression, the notation $S \in \mathbb{R}^{H \times W \times N_C \times D}$ is used only to specify the **shape** of the stacked tensor containing all saliency maps. Thus, these two notations serve different purposes (*composition* vs *shape),* and are not conflicting.
>
> However, we agree that the original phrasing may have caused confusion, and we have revised the paper to clarify this distinction more explicitly.
>
> **(b) Correlation map dimension**
>
> The reviewer assumes that a correlation map must be scalar-valued with shape $H\times W\times N_C$ which corresponds to **cosine similarity**.
>
> However, in correlation-based OVSS methods, including CAT-Seg and ESC-Net, the similarity maps are then projected to **multi-dimensional correlation maps via MLP** and we follow this common practice to represent the correlation maps as the shape $H\times W\times N_C\times D$. Our uploaded code implements this exact projection step, confirming the correctness of the shape.
>
> We have revised the paper to better describe this pipeline, including clearer explanations of the multi-channel correlation representation.
>
> > **Overly Convoluted and "Chaotic" Pipeline.**
>
> **We thank the reviewer for this careful observation and we would like to clarify our motivation.** We elaborate below:
>
> **(1) Correlation maps and saliency maps** serve **distinct and complementary roles.**
>
> Correlation contains **rich cross-modal correlation features**, which our SDM uses for semantic localization. Saliency provides a **scalar importance score** for disentanglement. It captures *where* meaningful evidence appears, while correlation encodes *what* semantic cues that region lies in. Using both is necessary for stable separation in SDM. The gradient-based saliency generation is not arbitrary but essential for robust disentanglement.
>
> **(2) Attention maps alone are insufficient because they are not causal.**
>
> Attention maps are often diffuse and may highlight irrelevant regions, as shown in prior OVSS works (e.g., PnP-OVSS). They do not indicate *which pixels causally contribute* to the similarity score. Gradient-based saliency gives a **causal and class-discriminative signal**, allowing SDM to reliably identify informative vs contextual regions.
>
> **(3) Gradient-based saliency is standard practice in OVSS pipelines.**
>
> Recent OVSS methods, including PnP-OVSS and IteRPrimE, leverage GradCAM-style gradients to obtain fine-grained class-specific saliency. Our design directly follows this widely adopted approach.
>
> **(4) Efficiency and other saliency generation approach**
>
> Our model complexity analysis proves that our model achieves least latency and comparable parameter sizes and GFLOPs. Our work reveals the potential of leveraging saliency for the ensemble foreground/background disentanglement. Although other saliency generation approaches exist, systematically comparing them is beyond our present scope and may be an interesting future direction.
>
> We improved Section 3 to explicitly describe the roles of correlation and saliency, the necessity of gradient-based sharpening, and how the components interact. We appreciate the reviewer for pointing this out, and we believe the revised explanation resolves the confusion.
>
> > **Marginal Ablation Gains.**
>
> The Variant I corresponds to SOTA model baseline results of CAT-Seg (12.0, 19.0, 31.8, 57.5, 94.6, 77.3) and does **not represent** our ablation baseline (DiSa without SDM) in top section of Table 2. To better show the impact of SDM, we replaced this line by our ablation baseline. Compared to our baseline results (IV) in the bottom section row, (VI) improves +0.1, +0.4, +0.8, +0.2, +0.1, -0.2 which is +0.23 on average which is meaningful for our one of three lightweight yet effective designs in HRM. We understand the reviewer’s confusion and appreciate this helpful suggestion.
>
> In summary, we have made these points clearer in the revised manuscript. We appreciate your valuable suggestions and hope that our responses have addressed your concerns and clarified the contributions of our work. We look forward to your responses soon.
>
> Best regards,
>
> Authors

---

> ### Author Response · Authors · 2025-11-26
> **Looking Forward to Further Discussions with Reviewer RfyL**
>
> Thank you for your appreciation of our work. In the response, we have clarified and addressed several misunderstandings like contradictory notions and chaotic pipeline. Given the provided suggestions, we also corrected several typos like Figure 6 and clarified the Variant I baseline in Table 2. We hope the revision paper has addressed your concerns and captured your interest. We look forward to engaging with you further.

---

> > ### Comment · Reviewer_RfyL · 2025-11-27
> >
> > Thank the authors for the detailed response, I have updated my overall rating and confidence score accordingly.

---

> > > ### Author Response · Authors · 2025-11-27
> > > **Official Comment by Authors**
> > >
> > > Thank you for your thoughtful response and we truly appreciate the time you spent reviewing our rebuttal. If any remaining points still require clarification, we would be happy to provide additional details. We hope that our responses have addressed your concerns sufficiently and that our contributions are reflected accordingly in your final evaluation.
> > >
> > > Thank you again for your time and consideration.

---

### Official Review · Reviewer_hWwH · 2025-11-01

**Soundness:** 3
**Presentation:** 3
**Contribution:** 2
**Rating:** 4
**Confidence:** 4

**Summary:**

The paper introduces DiSa, a novel saliency-aware foreground-background disentangled framework for open-vocabulary segmentation. DiSa introduces a Hierarchical Refinement Module (HRM) that captures spatial context through Pixel-, Category-, and Semantic-wise Refinement. DiSa achieves performance gains over six benchmarks.

**Strengths:**

1. The paper is clearly written.
2. The idea is novel and make senses to me.
3. The modular design can easily slot into CLIP-based baselines.

**Weaknesses:**

1. The saliency is derived from cross-attention + Grad-CAM reweighting via an auxiliary ITM loss tied to segmentation supervision. It may introduce label leakage.
2. The performance gains are limited from Table1.
3. The foreground/background Token Selection is a bit ambiguous. It is better to elaborate more. And is it possible to use the model to do foreground/background segmentation as well?

**Questions:**

1. How do you select top-k visual tokens in the correlation maps?
2. What happens if you freeze CLIP completely (no key/query projection tuning) to preserve open-vocabulary behavior?
3. It would be better if you could show some failure cases.

---

> ### Author Response · Authors · 2025-11-20
> **(1/2) Official Comment by Authors**
>
> We sincerely thank the reviewer for the thoughtful feedback and recognizing the technical merit of our work. We appreciate that the reviewer thinks our paper is novel, has clear motivation, and can be adapted to existing works. Below we address each of the concern in detail:
>
> > Data leakage.
>
> We thank the reviewer for raising this concern. We clarify that **our saliency generation does *not* introduce label leakage**, because **ground truth is *never* involved in saliency computation during inference.**
>
> **(1) Ground truth is *not* used in inference.**
>
> At inference, our model uses:
>
> - the regressed vector of image-text matching pair which is $y^{itm}$ in Eq. 1.
>
> **No segmentation mask, class label, or pixel-level supervision is needed during inference.**
>
> **We appreciate the reviewer’s comment and have revised the paper to explicitly clarify that ITM and all segmentation supervision are used only during training, while inference relies solely on image–text alignment.**
>
> > Limited gain.
>
> We respectfully disagree with the claim that improvements are limited.
>
> In semantic segmentation:
>
> - **+1.0 mIoU** is typically considered a *significant* gain.
>
> For example, the CVPR 2025 paper DPSeg reports improvements of
>
> - **+0, +0.5, +1.1, +0.6, +1.4 mIoU (average +0.72 mIoU)**
>
> Under this context, DiSa achieves:
>
> - **+0.6, +0.8, +0.8, +1.2, +1.0 mIoU (average +0.88 mIoU)**
>
>     which are **clearly substantial** and well beyond typical incremental improvements.
>
>
> Meanwhile, given the zero-shot nature of OVSS and challenging task setting (e.g., testing set with more than 800 classes), the average performance gain of +1.2 mIoU over six benchmarks under both base- and large-scale VLMs are promising.
>
> > FG/BG selection clarification.
> >
>
> We thank the reviewer for the thoughtful comments on our core design. We clarify that our method is **not designed** to predict strict FG/BG segmentation. Instead, our model leverages implicit FG/BG definition using saliency to conduct explicit disentanglement **of feature token**.
>
> **(1) Our contribution: saliency-guided semantic disentanglement**
>
> The goal of our token selection is to separate **salient regions** that contain rich semantic and structural cues from **non-salient regions** that provide contextual and environmental information for each class. This explicit disentanglement allows the two branches to focus on *complementary* aspects of the same category:
>
> - **Foreground branch:** captures semantic rich, informative details (textures, edges, structures).
> - **Background branch:** captures contextual cues (surroundings, contextual patterns).
>
> This design is not intended to output a standalone FG/BG mask but to **improve segmentation by modeling intra-class semantic diversity**, which we show to be effective across datasets.
>
> **(2) We intentionally do *not* impose a strict FG/BG boundary**
>
> We avoid hard binary FG/BG segmentation for two reasons:
>
> - There is **no universally agreed definition** of foreground vs background, especially in open-vocabulary settings (e.g., “road,” “sky,” “wall,” “grass” are stuff but may be salient or non-salient depending on the image).
> - Many objects include both salient and non-salient regions (e.g., part of the wall with texture vs flat wall), and imposing a strict binary split would be arbitrary and harmful.
>
> Our soft token-level saliency-based selection reflects **semantic contribution**, which matches the nature of CLIP/VLM attention.
>
> **(3) This soft separation is a key source of robustness**
>
> Because the FG/BG split is *not* strict, the two branches act like an **ensemble** that captures complementary cues. Even if part of the salient region is misassigned:
>
> - the FG branch still captures strong local semantics,
> - the BG branch captures global and contextual patterns,
>
> As shown in Section E of our updated appendix, performance remains stable even when some salient regions are misassigned, confirming that the model is **robust precisely because the split is intentionally soft and not treated as a binary segmentation task.**
>
> Thanks for this important question. We have revised the introduction and method sections to highlight the motivation of our token selection mechanism.

---

> ### Author Response · Authors · 2025-11-20
> **(2/2) Follow-up of Previous Comment**
>
> > top-k token selection.
>
> Thanks for raising this question. In our method, **foreground correlation tokens are selected based on the top-k saliency scores of the saliency map**. The procedure is straightforward:
>
> 1. After we obtain the saliency map $S$ for each class, we have a **saliency score** for each spatial location (x, y).
> 2. We then **rank all locations by their saliency scores** and simply select the **top-k locations.**
> 3. The correlation tokens at these selected positions are selected as foreground, salient regions and fed into the foreground branch; all remaining tokens are fed into the background branch.
>
> **We appreciate the reviewer for pointing this out and helping us improve the clarity of our token-selection description.**
>
> > Frozen CLIP.
>
> We thank the reviewer for this insightful comment. In our implementation, we follow CAT-Seg to fine-tune the Queries/Values of CLIP encoders. CAT-Seg reports that fine-tuning consistently outperforms frozen CLIP, and among different tuning strategies, QV fine-tuning yields the best performance. In our preliminary experiments, we also observe the same trend where fine-tuning noticeably improves performance vs frozen. Intuitively, this is because fine-tuning allows the model to adapt CLIP features to segmentation-specific cues required for dense prediction, while still preserving its open-vocabulary behavior.
>
> > Failure cases.
>
> We thank the reviewer for this important suggestion and we have added failure case analysis with visualizations in Section E in the appendix. It shows an example of foreground regions misinterpreted as non-salient because k is not enough for crowded scenes. However, our ensemble design of dual branches enables robustness by capturing preserving complementary cues explicitly.
>
> Once again, thank the reviewer for these constructive questions and feedback. The comments have significantly improved the clarity and impact of our work. We hope our responses adequately address all concerns and look forward to further responses.
>
> Best regards,
>
> Authors

---

> > ### Comment · Reviewer_hWwH · 2025-11-28
> >
> > Thank you for the response. Some of my concerns have been addressed. However, I remain a bit confused about the use of salient regions. For example, in Figure 5, is the foreground specifically designed to highlight the cows? How exactly is the “foreground” defined in this context? Does it also contain persons? In Figure 6, even though the foreground map is not very precise, the prediction is still correct. Could you clarify the role that the foreground salient region plays here? How sensitive is your method to inaccuracies in the foreground extraction?

---

> > > ### Author Response · Authors · 2025-11-28
> > > **Follow-up Response to Reviewer hWwH’s Question on Saliency**
> > >
> > > We appreciate the reviewer’s follow-up questions and are happy to clarify the role and definition of salient regions in our framework.
> > >
> > > **(1) Definition of “foreground”**
> > >
> > > Unlike previous saliency-related works, we propose to leverage a class-level saliency information. In our observations in ablation studies, this atomic disentanglement enhances the fine-grained understanding of each class.
> > >
> > > Our method follows the current correlation-based paradigm in OVSS, where class-specific correlation maps are generated for every category. To address the foreground bias problem, we use saliency maps to explicitly split each class’s tokens into *foreground* and *background* subsets. Note that we obtain $N_C$ saliency maps and they are focusing on the salient, foreground regions of each class. Our designed dual-branch design encourages the two sets to learn complementary but distinct cues.
> > >
> > > **(2) Figure 5**
> > >
> > > Fig. 5 visualizes the foreground tokens for the “cow” class only, therefore, it does not contain “person”. Each class has its own foreground map (e.g., the “person” class also has person-specific salient regions).
> > >
> > > **(3) Sensitivity**
> > >
> > > Although the saliency map is imprecise from human perspective in Fig. 6, the class-specific saliency provides fine-grained cues (most semantically informative details such as texture and local structures) that complement the class correlation maps. This example is aligned with claims of prior works [1-2] that saliency focuses on small-scale details from different objects of the same class without identifying the entire regions occupied, yet segmentation remains robust because our dual branches ensemble unique attributes of explicit disentangled regions.
> > >
> > > Thank you again for offering this opportunity to explain our novel design in detail and look forward to further discussions.
> > >
> > > [1] Reducing Information Bottleneck for Weakly Supervised Semantic Segmentation
> > >
> > > [2] Extracting Class Activation Maps from Non-Discriminative Features as well

---

> ### Author Response · Authors · 2025-11-26
> **Looking Forward to Further Discussions with Reviewer hWwH**
>
> Thank you for your time and attention. In the response, we have clarified and addressed several misunderstandings like data leakage and limited performance gain. We also provided failure cases which further demonstrated our design's robustness. We hope you have had the opportunity to review our revisions and look forward to further engaging discussions with you.

---

### Official Review · Reviewer_WaQ7 · 2025-11-01

**Soundness:** 3
**Presentation:** 2
**Contribution:** 3
**Rating:** 6
**Confidence:** 4

**Summary:**

This paper effectively addresses the foreground bias in Vision-Language Models (VLMs) for open-vocabulary semantic segmentation by introducing a saliency map module to decouple foreground and background regions. The authors further propose a Hierarchical Boundary Modulation (HBM) module that refines the segmentation output at the pixel, semantic, and category levels. Their method also achieves State-of-the-Art (SOTA) performance on multiple official benchmarks.

**Strengths:**

1. The fundamental problem is well-identified and critical: The authors pinpoint the crucial issue of Vision-Language Models (VLMs) being inherently biased toward foreground objects in dense prediction tasks like semantic segmentation. Their proposed saliency map-based method provides an elegant and direct solution for decoupling foreground and background representations.

2. The empirical results are well-validated: The model not only achieves State-of-the-Art (SOTA) performance but demonstrates robustness and superior generalization by outperforming competitors across all official benchmarks tested.

**Weaknesses:**

1. The proposed module to extract foreground/background region is based GradCAM. While maintaining a comparable GFLOP count, the reliance on a gradient-based method like GradCAM may lead to a slower inference speed due to the required backward pass.

**Questions:**

1. Please check and correct the typo on Line 245: "re-weigh" should be reviewed for proper hyphenation or spelling (e.g., "reweight" or "re-weight").

2. Does  Variant I in Table 5 represent the performance of the DiSa model without the Saliency Boundary Module (SBM)?

3. For 'stuff' classes (e.g., wall or sky), what precisely does the saliency map capture? Is the definition of saliency in this work consistent with prior literature, or does it merely indicate a region of higher model confidence?

---

> ### Author Response · Authors · 2025-11-20
> **Official Comment by Authors**
>
> We sincerely thank the reviewer for the thoughtful and detailed feedback. We appreciate that the reviewer thinks our paper is well-identified, critical, and well-validated. We provide clarifications and evidence that address each of the raised concerns in detail below:
>
> > **Slower inference speed.**
> >
>
> We thank the reviewer for raising this point.
>
> **(1)** We clarify that our gradient-based computation is *lightweight* because the gradient is **only backpropagated to the attention maps**, not through the entire network. This is a very short computational path. As a result, the overhead of this saliency process is minimal.
>
> **(2)** To validate this claim, we include inference latency in Table 1 below. Our model achieves **lower latency** to SOTA methods. This confirms that our model is lightweight yet powerful.
>
> Table 1: Model complexity comparison
>
> |  | # Params (M) | GFLOPs | Inference time (s) |
> | --- | --- | --- | --- |
> | ZegFormer | 531.2 | 19,425.6  | 3.11 |
> | ZSseg | 530.8 | 22,302.1 | 3,10 |
> | OVSeg | 532.6 | 19,345.6 | 2.98 |
> | CAT-Seg | 433.7 | 2,121.1 | 0.78 |
> | ESC-Net | 451.3 | 2,203.5 | 0.76 |
> | Ours | 456.2 | 2,287.3 | 0.69 |
>
> > Questions
> >
>
> We thank the reviewer for highlighting these questions and pointing out one typo. We clarify each of them below in detail:
>
> (1) Variant I in the Table
>
> This line corresponds to SOTA model baseline results of CAT-Seg and does **not represent** our ablation baseline. To better show the impact of SDM, we replaced this line by our ablation baseline in top section (DiSa without SDM). We understand the reviewer’s confusion and appreciate this helpful suggestion.
> (2)  Saliency map of ‘stuff’ classes
>
> The saliency maps, for *stuff* classes, highlights **the most semantically informative or attribute-rich regions**. This behavior aligns with our core motivation: *even within the same semantic category, different regions may contribute differently to the textual concept.* For example, in classes like wall and sky, the textured parts of a wall and the cloud structures in the sky provide relatively stronger visual cues, while others serve as contextual or peripheral background.
>
> This definition is consistent with prior work that uses saliency to reflect **semantic contribution**, but with a slight modification: we obtain saliency maps for each class instead of one single saliency map for all classes. In our method, saliency therefore represents regions of semantic details and informative structures, not merely regions of model confidence.
>
> We appreciate the reviewer’s insights, which align closely with our motivations and observations, and we believe the additional clarifications and experiments directly address the concerns raised. Look forward to the further responses.
>
> Sincerely,
>
> Authors

---

> > ### Comment · Reviewer_WaQ7 · 2025-11-24
> > **No further problem**
> >
> > Thank the authors for their clarifications. I have no further concerns.

---

> ### Author Response · Authors · 2025-11-24
> **Grateful Acknowledgment and Commitment to Advancing Our Work**
>
> Thank you for your effort and encouraging feedback in reviewing! We appreciate your understanding and recognition of our efforts in advancing this work. Your acknowledgment inspires us to continue contribute to the development of the OVSS field.
>
> Thank you again for maintaining your positive score, and we sincerely wish you all the best as well!

---

### Official Review · Reviewer_xghB · 2025-11-01

**Soundness:** 3
**Presentation:** 3
**Contribution:** 3
**Rating:** 6
**Confidence:** 3

**Summary:**

The paper proposes DiSa, a saliency-aware method for open-vocabulary segmentation that disentangles foreground-background using saliency for decomposition and hierarchical feature refinement. The key contributions are:
- Thew saliency aware disentanglement module that uses saliency for adaptive foreground to background separation to mitigate foreground bias.
- Hierarchical Refinement Module that is used to capture detailed spatial and channel context in 3 stages (Pixel, category, Semantic) to improve limited spatial localization.
- SOTA performance across six large open vocabulary semantic segmentation benchmarks.

**Strengths:**

- Both the DSM and HRM are shown to effectively mitigate their issues and the ablation studies show a strong motivation for the current elements in the method.
- Good experimental results and the increase in PAS-20b indicates the results are supported by the theory.
- Good performance with no additional datasets and low computational cost (GFLOPs).

**Weaknesses:**

- The entire disentanglement pipeline is dependent on the quality and accuracy of the ITM loss. If it fails to localize the object for novel classes the split will be flawed.
- While GFLOPs are low the pipeline is complex which can introduce a higher training overhead and fragility (more failure points).
- k=96 is a fixed value. This doesn't account for object that vary widely in size or partially visible which can impact performance on diverse scenes.

**Questions:**

- To isolate the novelty of the SDM what is the performance gap between using $L_{itm}$ gradient cue vs a simpler method like CLIP attention maps with the dual branch structure?
- How does the weighted feature aggregation block compare to simple concatenation or element-wise summation?
- The authors should provide an analysis of failure cases and visualizations where the disentanglement fails (e.g. background regions misclassified as foreground or ambiguous boundaries) to illustrate the limitations of the SDM.

---

> ### Author Response · Authors · 2025-11-20
> **Official Comment by Authors**
>
> We sincerely thank the reviewer for the thoughtful feedback. We are encouraged by the appreciation of our motivation and comprehensive results. We provide clarifications and evidence that address each concern below:
>
> > Dependence on ITM loss.
> >
>
> **We thank the reviewer for the insightful comment. We clarify that our method does *not* depend strongly on the quality of the ITM loss:**
>
> (1) The ITM is used **only as an auxiliary sharpening loss** to enhance contrast and to back-propagate localized gradients for saliency generation. It is *not* the primary signal that we rely on for producing the FG/BG split.
>
> (2) The disentanglement in SDM indeed relies on **image-text attention maps**, which are standard in OVSS and provide **robust localization cues**. The attention maps, rather than ITM, are the core driver of token separation. Importantly, the attention maps are **continuously optimized during training**, which stabilizes the quality even if the ITM gradients are imperfect.
>
> (3) Our design is robust to imperfect separations. The dual-branch architecture serves as an **explicit ensemble**, similar to mixture-of-experts (MOE) or multi-head attention: each branch specializes on complementary feature subsets. This structure **mitigates errors from imprecise FG/BG separation**, because the model is not forced to have a strict boundary. In Section E of our updated appendix, we provide visualization showing that even when separation is imprecise, the final predictions can still be optimal.
>
> **We have revised the main text to make this robustness and the auxiliary nature of the ITM loss clearer.**
>
> > Training overhead.
> >
>
> We appreciate the reviewer’s concern regarding training cost and fragility.
>
> (1) We provide training cost comparison with CAT-Seg in the Table 1 below, showing our model has comparable costs to CAT-Seg under the same setting.
>
> Table 1: Training overhead (hr) comparison
>
> |  | CLIP ViT-B | CLIP ViT-L |
> | --- | --- | --- |
> | CAT-Seg | 19.13 | 26.04 |
> | DiSa | 21.40 | 28.83 |
>
> (2) Regarding fragility, several modules enable robustness. The **dual-branch disentanglement** captures ensemble, complementary cues even when the separation is not perfect. Moreover, the auxiliary **ITM loss** and **HRM** further improve precision.
>
> > Fixed value of k.
> >
>
> We thank the reviewer for raising this concern. Table 3 already includes an **ablation of various k**, which shows that performance is stable within a broad range (±0.5 mIoU). This is because, in most real-world scenes, FG tokens seldom occupy extreme proportions, thus, moderate shifts in k do not significantly affect token partitioning. As discussed above, our design tolerates imprecise separation since it is not strictly optimized for FG/BG segmentation. Instead, our ensemble design enables robustness of k.
>
> We agree that **dynamic or adaptive token selection** is an interesting future direction, and we verify that dynamic k pre-calculated for each benchmark improve the performance.  However, this is out of our scope and we have added this to the discussion of future works.
>
> > Ablation studies in questions.
> >
>
> We appreciate these excellent suggestions and we clarify each in detail below:
>
> **(1) Ablation of gradient**
>
> In our method, the saliency map is produced by **backpropagating the ITM loss**, similar to Grad-CAM but adapted to the image-text alignment module. This gradient-based map is required and our motivation lies in the foreground (salient) and background (non-salient) separation.
>
> **We have emphasized in the paper that ITM gradients are required for differentiability and saliency generation**, while CLIP attention cannot substitute for gradient-based saliency.
>
> **(2) Ablation of aggregation.**
>
> For feature aggregation, we conduct additional experiments that compare our weighted aggregation to Attention-based and Hard aggregation **in Table 2**. Our design achieves significant improvements, indicating that our design is robust to various aggregation selections.
>
> Table 2: Ablation study of aggregation
>
> |  | A-847 | PC-459 | A-150 | PC-59 | PAS-20 | PAS-20b |
> | --- | --- | --- | --- | --- | --- | --- |
> | Attn agg | 12.5 | 20.3 | 33.5 | 59.0 | 97.3 | 79.4 |
> | Hard agg | 12.2 | 19.9 | 32.9 | 58.7 | 96.4 | 80.1 |
> | Weighted agg (ours) | 12.6 | 20.3 | 33.7 | 59.3 | 97.0 | 79.9 |
>
> **(3) Failure case analysis.**
>
> To illustrate the limitations, we have added  failure case with visualizations in Section E in the appendix. It shows an example of foreground regions misinterpreted as non-salient because k is not enough for crowded scenes. However, our ensemble design of dual branches enables robustness by capturing preserving complementary cues explicitly.
>
> We thank the reviewer again for raising these aspects, which help us include clearer clarification and analyses. We hope our responses address the concerns comprehensively and look forward to further response.
>
> Sincerely,
>
> Authors

---

> ### Author Response · Authors · 2025-11-26
> **Looking Forward to Further Discussions with Reviewer xghB**
>
> Thank you for your valuable suggestions. In the response, we have incorporated additional training overhead analysis and ablation studies. We have also improved the clarifications of the dependence on ITM loss. Furthermore, we provided failure cases which further demonstrated our design's robustness. We hope you have had the chance to review these updates and look forward to engaging in further discussions with you.

---

### Author Response · Authors · 2025-11-25
**Summary and Answers of Official Reviews (1/2)**

Dear AC and Reviewers,

We sincerely thank all reviewers for their constructive comments and insights. We appreciate the recognition of our paper and the valuable suggestions for enhancing the quality. Below, we summarize the key strengths of our paper by reviewers in Table 1. These acknowledgements motivate us to further improve the quality and impact of our work.

Table 1: Recognized Contributions.

| **Contribution** | **Reviewer** | **Official Review** |
| --- | --- | --- |
| **1. Clear Motivation** | xghB | "…show a strong **motivation** for the current elements in the method." |
|  | WaQ7 | "The fundamental problem is **well-identified**..." |
|  | RfyL | "The paper's primary strength lies in its **clear and accurate motivation**." |
| **2. Identification of Current Works’ Limitations** | xghB | "…to mitigate **foreground bias**." “…to improve **limited spatial localization**.’ |
|  | WaQ7 | "This paper effectively addresses the foreground bias in Vision-Language Models (VLMs)…" |
|  | RfyL | "The **identification** of "Foreground Bias" and "Limited Spatial Localization" as two distinct, critical failures of VLMs…" |
| **3. Novel and Interesting Idea** | xghB | "The **key contributions** are: Thew saliency aware disentanglement module…Hierarchical Refinement Module…" |
|  | WaQ7 | " Their proposed saliency map-based method provides an **elegant and direct solution** for decoupling foreground and background representations." |
|  | hWwH | "The idea is **novel** and make senses to me." |
| **4. Strong Performance Improvement** | xghB | "**SOTA** performance across six large open vocabulary semantic segmentation benchmarks." |
|  | WaQ7 | “The empirical results are **well-validated**...” |
|  | hWwH | “ DiSa achieves performance **gains** over six benchmarks.” |
|  | RfyL | “…bring additional **performance improvement** over baseline.” |
| **5. Robustness and Generalization** | xghB | “Good performance with **no** additional datasets and **low** computational cost (GFLOPs).” |
|  | WaQ7 | “…demonstrates **robustness and superior generalization**.” |
|  | hWwH | “The modular design can **easily slot** into CLIP-based baselines.” |

Additionally, reviewers **WaQ7** and **RfyL** confirmed that all their concerns were resolved, and reviewer **RfyL** decided to **raise the final ratings while reducing the confidence score to 2**. Reviewer **hWwH** acknowledged that most concerns were addressed and sought require clarifications on the role and definition of saliency, which we believe our latest explanation has fully resolved.

Thank you again for your support and suggestions, which have greatly improved the quality of our paper.

Best regards,

Authors

---

### Author Response · Authors · 2025-11-25
**Summary and Answers of Official Reviews (2/2)**

Thank all the reviewers for the insightful questions and careful review which help us to improve the quality of our work. Here we answer the questions of common concern.

### **1. Pipeline Complexity**

We clarify that our gradient-based computation is *lightweight* because the gradient is **only backpropagated to the attention maps**, not through the entire network. This is a very short computational path. To validate this claim, we include inference latency in Table 1 below. Our model achieves **lower latency** to SOTA methods. This confirms that our model is lightweight yet powerful.
Table 1: Model complexity comparison

|  | # Params (M) | GFLOPS | Inference time (s) |
| --- | --- | --- | --- |
| ZegFormer | 531.2 | 19,425.6  | 3.11 |
| ZSseg | 530.8 | 22,302.1 | 3,10 |
| OVSeg | 532.6 | 19,345.6 | 2.98 |
| CAT-Seg | 433.7 | 2,121.1 | 0.78 |
| ESC-Net | 451.3 | 2,203.5 | 0.76 |
| Ours | 456.2 | 2,287.3 | 0.69 |

### **2. Potential Label Leakage and ITM Loss Reliability**

**Ground truth is *never* involved in saliency computation during inference.** At inference, our model uses the regressed one hot vector of image-text matching pair which is $y^{itm}$ in Eq. 1. **No segmentation mask, class label, or pixel-level supervision is needed during inference.**

Besides, in terms of the reliability:

(1) The ITM is used only as an **auxiliary** sharpening loss to enhance contrast and to back-propagate localized gradients for saliency generation. The disentanglement in SDM indeed relies on **image-text attention maps**, which are standard in OVSS and provide **robust localization cues**. Importantly, the attention maps are **continuously optimized during training**, which stabilizes the quality even if the ITM gradients are imperfect.

(3) Our design is robust to imperfect separations. The dual-branch architecture serves as an **explicit ensemble**, similar to mixture-of-experts (MOE) or multi-head attention: each branch specializes on complementary feature subsets. This structure **mitigates errors from imprecise FG/BG separation**, because the model is not forced to have a strict boundary. In Section E of our updated appendix, we provide visualization showing that even when separation is imprecise, the final predictions can still be optimal.

### **3. Variant I (Baseline) results in Table 2**

The Variant I corresponds to SOTA model baseline results of CAT-Seg (12.0, 19.0, 31.8, 57.5, 94.6, 77.3) and does **not represent** our ablation baseline (DiSa without SDM) in top section of Table 2. To better show the impact of SDM, we replaced this line by our ablation baseline. Compared to our baseline results (IV) in the bottom section row, (VI) improves +0.1, +0.4, +0.8, +0.2, +0.1, -0.2 which is +0.23 on average which is meaningful for our one of three lightweight yet effective designs in HRM. We understand the reviewer’s confusion and appreciate this helpful suggestion.

### **4. Failure Case Analysis**

We have added failure case analysis with visualizations in Section E in the appendix. It shows an example of foreground regions misinterpreted as non-salient because k is not enough for crowded scenes. However, our ensemble design of dual branches enables robustness by capturing preserving complementary cues explicitly.

---

### Meta-Review · Area_Chair_vpDc · 2026-01-07

**Summary:**

In the initial review round, the paper received mixed scores (6, 6, 4, 2), with reviewers raising substantial concerns. The primary issues were as follows:

Reviewer xghB: The method is complex and error-prone; lacks ablation studies on the hyperparameter k.

Reviewer WaQ7: Slow inference time.

Reviewer hWwH: Introduces potential label leakage; achieves only limited performance gains; provides unclear explanations.

Reviewer RfyL: Methodology and notations are unsound; pipeline is complex; ablation studies show only marginal gains.

Overall, the reviewers' concerns center on the method's complexity, risk of errors or label leakage, methodological soundness, limited empirical benefits, slow inference, and insufficient ablation or clarification.

**Reviewer Concerns:**

Reviewer xghB: The concern regarding pipeline complexity is not adequately addressed. While the authors provide parameter counts and GFLOPs, they do not sufficiently mitigate the issue of generating multiple intermediate maps, which contributes to overall complexity and potential error-proneness.

Reviewer WaQ7: The concern about slow inference time has been satisfactorily addressed.

Reviewer hWwH: The concern over limited performance gains is not well-resolved. I do not find the authors' claim that a 1-point mIoU improvement is significant to be convincing. Other concerns from this reviewer appear addressed.

Reviewer RfyL: The concern about unnecessary pipeline complexity remains unaddressed. Other concerns from this reviewer have been handled.

**Reviewer Scores:**

Reviewer xghB: will keep initial score (6)

Reviewer WaQ7: will keep initial score (6)

Reviewer hWwH: will keep initial score (4)

Reviewer RfyL: may increase score to 4

Overall, I think this is a borderline paper. I align with Reviewers xghB and RfyL that the proposed method is complex, making it prone to errors. I therefore recommend rejection. I encourage the authors to simplify the pipeline, strengthen the ablation studies, and provide clearer evidence of practical advantages before resubmitting a revised manuscript elsewhere.

---

### Decision · Program_Chairs · 2026-01-26

Reject